# Endoreversible Models for the Thermodynamics of Computing

**DOI:** 10.3390/e22060660

**Published:** 2020-06-15

**Authors:** Alexis De Vos

**Affiliations:** Vakgroep elektronika en informatiesystemen, Universiteit Gent, Technologiepark 126, B-9052 Gent, Belgium; alexis.devos@ugent.be

**Keywords:** macroentropy, microentropy, endoreversible engine, reversible computing, Landauer’s principle

## Abstract

Landauer’s principle says that, in principle, a computation can be performed without consumption of work, provided no information is erased during the computational process. This principle can be introduced into endoreversible models of thermodynamics.

## 1. Introduction

Thermodynamics of computing has a peculiar history. Many years, scientists have searched for the minimum energy needed to perform an elementary computing step. It was Landauer [1,2,3,4] who demonstrated in the period 1960–1990 that, in principle, computing can be performed without energy consumption, provided the computing process applies exclusively logically reversible computing steps. As long as information is not destroyed and computing is performed infinitely slowly, no work has to be supplied to the computer. Only erasure of information requires energy input. It is remarkable that we had to wait until the period 2012–2018 to have experimental confirmation [5,6,7,8] of Landauer’s principle.

Basic thermodynamics, i.e., the Carnot theory, describes thermal engines acting infinitely slowly. In 1975, Curzon and Ahlborn [9] presented a thermodynamical model for an engine working at non-zero speed: the endoreversible engine. It consists of a reversible core, performing the actual conversion (of heat into work) and two irreversible channels for the heat transport. The approach turned out to be very fruitful: not only processes in engineering, but also in physics, chemistry, economics, etc. can successfully profit from endoreversible modelling, especially when processes happen at non-zero speed and thus tasks are performed in a finite time [10,11].

The present paper is an attempt to apply the endoreversible scheme to the Landauer principle, thus to thermodynamically describe computing at a non-zero speed.

## 2. Logic Gates

Any computer is built from basic building blocks, called gates. In a conventional electronic computer, such building block is e.g., a not gate, an or gate, a nor gate, an and gate, a nand gate, etc. Such gate has both a short input (denoted with subscript 1) and a short output (denoted with subscript 2). As an example, Table 1a defines the and gate, by means of its truth table. We see an input word $A_{1}B_{1}$ and the corresponding output word $A_{2}$. If the input word is given, the table suffices to read what the output ‘will be’. If, however, the value of the output word is given, this information is not sufficient to recover what the input word ‘has been’. Indeed, output $A_{2}=0$ can equally well be the result of either $A_{1}B_{1}=00$ or $A_{1}B_{1}=01$, or $A_{1}B_{1}=10$. For this reason, we say that the gate is logically irreversible. In contrast, the not gate is logically reversible, as can be verified from its truth table in Table 1b. Indeed, knowledge of $A_{1}$ suffices to know $A_{2}$, but also: knowledge of $A_{2}$ suffices to know $A_{1}$.

We not only can distinguish logically irreversible gates from logically reversible gates. We also can quantify how strongly a gate is irreversible. For this purpose, we apply Shannon’s entropy:$$S=-k\sum q_{i}\log\left(q_{i}\right)\;,$$
where *k* is Boltzmann’s constant and $q_{i}$ is the probability that a word ABC… (either an input word $A_{1}B_{1}C_{1}\ldots$ or an output word $A_{2}B_{2}C_{2}\ldots$) has a particular value. As an example, we examine Table 1a in detail. Let ${\left(q_{00}\right)}_{1}$ be the probability that input word $A_{1}B_{1}$ equals 00, let ${\left(q_{01}\right)}_{1}$ be the probability that $A_{1}B_{1}$ equals 01, let ${\left(q_{10}\right)}_{1}$ be the probability that it equals 10, and let ${\left(q_{11}\right)}_{1}$ be the probability that it equals 11. We, of course, assume $0\leq {\left(q_{i}\right)}_{1}\leq 1$ for all *i*, as well as $\sum {\left(q_{i}\right)}_{1}=1$. Let ${\left(q_{0}\right)}_{2}$ be the probability that output word $A_{2}$ equals 0 and let ${\left(q_{1}\right)}_{2}$ be the probability that $A_{2}$ equals 1. Inspection of Table 1a reveals that
$$\begin{array}{ccc} {\left(q_{0}\right)}_{2} & = & {\left(q_{00}\right)}_{1}+{\left(q_{01}\right)}_{1}+{\left(q_{10}\right)}_{1}\\ {\left(q_{1}\right)}_{2} & = & {\left(q_{11}\right)}_{1}\;. \end{array}$$

Automatically, we have $0\leq {\left(q_{i}\right)}_{2}\leq 1$ for both *i*, as well as $\sum {\left(q_{i}\right)}_{2}=1$. We now compare the entropies of input and output:$$\begin{array}{ccc} S_{1} & = & -k\sum {\left(q_{i}\right)}_{1}\;\log\left({\left(q_{i}\right)}_{1}\right)\\ S_{2} & = & -k\sum {\left(q_{i}\right)}_{2}\;\log\left({\left(q_{i}\right)}_{2}\right)\;. \end{array}$$

We find that these two quantities are not necessarily equal. For example, if the inputs 00, 01, 10, and 11 are equally probable, i.e., if
$${\left(q_{00}\right)}_{1}={\left(q_{01}\right)}_{1}={\left(q_{10}\right)}_{1}={\left(q_{11}\right)}_{1}=1/4\;,$$
then we have
$${\left(q_{0}\right)}_{2}=3/4\;\mathrm{and}\;{\left(q_{1}\right)}_{2}=1/4\;,$$
such that
$$\begin{array}{ccc} S_{1} & = & 2\;k\log(2)=2\;b\\ S_{2} & = & \left[\;2-\frac{3\log(3)}{4\log(2)}\;\right]\;k\log\left(2\right)\approx 0.811\;b\;, \end{array}$$
where b=klog(2) is called ‘one bit of information’. Thus, evolving from input to output is accompanied by a loss of entropy $S_{1}-S_{2}$ of about 1.189 bits. A similar examination of Table 1b leads to $S_{1}=S_{2}=1\;b$. Thus, both input and output contain one bit of information. There is no change in entropy: $S_{1}-S_{2}=0$.

A reversible computer is a computer exclusively built from reversible logic gates [12,13]. As among the conventional logic gates, only the not gate is logically reversible, we need to introduce unconventional reversible gates, in order to be able to build a general-purpose reversible computer. Table 2 shows two examples: the controlled not gate (a.k.a. the Feynman gate) and the controlled controlled not gate (a.k.a. the Toffoli gate). The truth table of the controlled not gate has the following properties:$$\begin{array}{ccc} {\left(q_{00}\right)}_{2} & = & {\left(q_{00}\right)}_{1}\\ {\left(q_{01}\right)}_{2} & = & {\left(q_{01}\right)}_{1}\\ {\left(q_{10}\right)}_{2} & = & {\left(q_{11}\right)}_{1}\\ {\left(q_{11}\right)}_{2} & = & {\left(q_{10}\right)}_{1}\;, \end{array}$$
such that $\sum \;{\left(q_{i}\right)}_{2}\;\log\left({\left(q_{i}\right)}_{2}\right)=\sum \;{\left(q_{i}\right)}_{1}\;\log\left({\left(q_{i}\right)}_{1}\right)$ and thus $S_{2}=S_{1}$. This result is true whatever the values of the input probabilities ${\left(q_{00}\right)}_{1}$, ${\left(q_{01}\right)}_{1}$, ${\left(q_{10}\right)}_{1}$, and ${\left(q_{11}\right)}_{1}$, thus not only if these four numbers all are equal to 1/4. The reason of this property is clear: the output words $A_{2}B_{2}$ of Table 2a are merely a permutation of the input words $A_{1}B_{1}$. Analogously, in Table 2b, the output words $A_{2}B_{2}C_{2}$ form a permutation of the input words $A_{1}B_{1}C_{1}$. Therefore, the controlled controlled not gate also satisfies $S_{2}=S_{1}$ and hence is logically reversible.

Figure 1 shows a c-MOS (i.e., complementary metal–oxide–semiconductor) implementation of these two reversible gates in a silicon chip.

The reader may easily verify that, in general, we are allowed to summarize as follows:if the logic gate is logically reversible, then entropy is neither increased nor decreased;if the logic gate is logically irreversible, then entropy is decreased.

Of course, in the framework of the second law, any entropy decrease sounds highly suspicious. The next section will demonstrate that fortunately there is no need to worry.

## 3. Macroentropy and Microentropy

Let the phase space of a system be divided into *N* parts. Let $p_{m}$ be the probability that the system finds itself in part # *m* of the phase space. Then, the entropy of the system is
(1)$$\sigma =-\;k\sum_{m=1}^{N}\;p_{m}\log\left(p_{m}\right)\;.$$

Figure 2a shows an example with N=15.

We now assume that the division of phase space happens in two steps. First, we divide it into *M* large parts (with M≪N), called macroparts. Then, we divide each macropart into microparts: macropart # 1 into $n_{1}$ microparts, macropart # 2 into $n_{2}$ microparts, ..., and macropart # *M* into $n_{M}$ microparts:$$n_{1}+n_{2}+\ldots +n_{M}=N\;.$$

We denote by $p_{i,j}$ the probability that the system is in microcell # *j* of macrocell # *i*. Let $q_{i}$ be the probability that the system finds itself in macropart # *i*:$$q_{i}=\sum_{j=1}^{n_{i}}\;p_{i,j}\;.$$

Figure 2b shows an example with N=15 and M=4 ($n_{1}=4$, $n_{2}=5$, $n_{3}=3$, and $n_{4}=3$); Figure 2c shows an example with N=15 and M=2 ($n_{1}=12$ and $n_{2}=3$).

Let σ be the entropy of the system consisting of the *M* macrocells. We have
(2)$$\sigma =-\;k\sum_{i=1}^{M}\;\sum_{j=1}^{n_{i}}\;p_{i,j}\log\left(p_{i,j}\right)\;.$$

One can easily check that this expression can be written as
(3)$$\sigma =-\;k\sum_{i=1}^{M}\;q_{i}\log\left(q_{i}\right)-\;k\sum_{i=1}^{M}\;q_{i}\sum_{j=1}^{n_{i}}\;\frac{p_{i,j}}{q_{i}}\;\log\left(\frac{p_{i,j}}{q_{i}}\right)\;.$$

The former contribution to the rhs of Equation (Equation 3) is called the macroentropy *S*, whereas the latter contribution is called the microentropy *s*. We identify the macroentropy with the information entropy of Section 2. We associate the microentropy with the heat *Q*, i.e., with the energy exchange which would occur, if the microentropy enters or leaves the system at temperature *T*, according to the Gibbs formula
$$s=\frac{Q}{T}\;.$$

Hence:$$\sigma =S+\frac{Q}{T}\;.$$

This decomposition can be expressed in several ways:$$\begin{array}{ccc} \mathrm{total entropy} & = & \mathrm{macroentropy}+\mathrm{microentropy}\\ & = & \mathrm{Shannon entropy}+\mathrm{Gibbs entropy}\\ & = & \mathrm{information entropy}+\mathrm{heat entropy}\;. \end{array}$$

We now assume that the probabilities of being in a particular microcell is the same in the three cases of Figure 2. For example, $p_{6}$ of Figure 2a equals both $p_{2,2}$ of Figure 2b and $p_{1,6}$ of Figure 2c. Then, Equations (Equation 1) and (Equation 2) tell us that the entropy σ is the same in the three cases:$$\sigma_{a}=\sigma_{b}=\sigma_{c}\;.$$

Assuming all probabilities $p_{i,j}$ are non-zero, it is clear that the macroentropies satisfy
$$0=S_{a}<S_{c}<S_{b}\;.$$

Therefore, the microentropies satisfy
$$\sigma =s_{a}>s_{c}>s_{b}\;.$$

In particular, the inequality $S_{c}<S_{b}$ corresponds with the inequality $S_{2}<S_{1}$ in Section 2 for the and gate in Table 1a. Indeed: Figure 2b corresponds with the left part of the truth table, whereas Figure 2c corresponds with the right part of the table. The decrease of macroentropy ($S_{2}<S_{1}$) thus is compensated by the increase of microentropy ($s_{2}>s_{1}$), leading to $\sigma_{2}=\sigma_{1}$, thus saving the second law: $\sigma_{2}\geq \sigma_{1}$.

## 4. Reversible Engine

Figure 3a is the classical model of the Carnot engine, consisting of
a heat reservoir at temperature $T_{1}$, providing a heat $Q_{1}$,a heat reservoir at temperature $T_{2}$, absorbing a heat $Q_{2}$, and a reversible convertor, generating the work *E*.

For our purpose, we provide each reservoir with a second parameter, i.e., the macroentropy *S*. See Figure 3b.

We write the two fundamental theorems of reversible thermodynamics:conservation of energy: the total energy leaving the convertor is zero:
$$-Q_{1}+E+Q_{2}=0\;;$$conservation of entropy: the total entropy leaving the convertor is zero:
$$-\left(\frac{Q_{1}}{T_{1}}+S_{1}\right)+0+\left(\frac{Q_{2}}{T_{2}}+S_{2}\right)=0\;.$$

Eliminating the variable $Q_{2}$ from the above two equations yields
(4)$$E=\left(1-\frac{T_{2}}{T_{1}}\right)\;Q_{1}+\left(S_{2}-S_{1}\right)\;T_{2}\;.$$

We can distinguish two special cases (Figure 4):If $S_{2}=S_{1}$, then we obtain from Equation (Equation 4) that
$$E=\left(1-\frac{T_{2}}{T_{1}}\right)\;Q_{1}\;,$$
known as Carnot’s law.If $T_{2}=T_{1}$ (say *T*), then we obtain from (Equation 4) that
$$E=(S_{2}-S_{1})\;T\;,$$
known as Landauer’s law.

We thus retrieve, besides Carnot’s formula, the priciple of Landauer: if no information is erased ($S_{2}=S_{1}$), then no work *E* is involved; if information is erased ($S_{2}<S_{1}$), then a negative work *E* is produced, meaning that we have to supply a positive work −E.

We note that, in Figure 4a, the arrows indicate the sence of positive $Q_{1}$ (heat leaving the upper heat reservoir) and positive $Q_{2}$ (heat entering the lower heat reservoir). Analogously, in Figure 4b, the arrows indicate the sence of positive $S_{1}$ (macroentropy leaving the upper memory register) and $S_{2}$ (macroentropy entering the lower memory register). In order to actually perform the computation in the positive direction, an external driving force is necessary. The next section introduces this ‘arrow of computation’.

## 5. Reversible Engine Revisited

Information is carried by particles. Therefore, we have to complement the reservoirs of Figure 3b with a third parameter, i.e., the chemical potential μ of the particles. See Figure 5. Besides a heat flow *Q*, a reservoir also provides (or absorbs) a matter flow *N*.

In conventional electronic computers, the particles are electrons and holes within silicon and copper. There, the particle flow *N* is (up to a constant) equal to the electric current *I*:N=I/q,
where *q* is the elementary charge. The chemical potential μ is (up to a constant) equal to the voltage *V*:μ=qV.

In the present model, we maintain the quantities *N* and μ, in order not to exclude unconventional computing, e.g., computation by means of ions, photons, Majorana fermions, ... or even good old abacus beads.

We write the three fundamental theorems of reversible thermodynamics:conservation of matter: the total amount of matter leaving the convertor is zero:
$$-N_{1}+0+N_{2}=0\;;$$conservation of energy: the total energy leaving the convertor is zero:
$$-\left(Q_{1}+\mu_{1}N_{1}\right)+E+\left(Q_{2}+\mu_{2}N_{2}\right)=0\;;$$conservation of entropy: the total entropy leaving the convertor is zero:
$$-\left(\frac{Q_{1}}{T_{1}}+S_{1}\right)+0+\left(\frac{Q_{2}}{T_{2}}+S_{2}\right)=0\;.$$

The first equation leads to $N_{2}=N_{1}$, which we simply denote by *N*. Eliminating the variable $Q_{2}$ from the remaining two equations yields the output work:$$E=\left(1-\frac{T_{2}}{T_{1}}\right)\;Q_{1}+\left(\mu_{1}-\mu_{2}\right)\;N+\left(S_{2}-S_{1}\right)\;T_{2}\;.$$

We can distinguish three special cases (Figure 6):If $\mu_{2}=\mu_{1}$ and $S_{2}=S_{1}$, then we obtain
$$E=\left(1-\frac{T_{2}}{T_{1}}\right)\;Q_{1}\;,$$
i.e., Carnot’s law.If $T_{2}=T_{1}$ and $S_{2}=S_{1}$, then we obtain
$$E=(\mu_{1}-\mu_{2})\;N\;,$$
known as Gibb’s law.If $T_{2}=T_{1}$ (say *T*) and $\mu_{2}=\mu_{1}$, then we obtain
$$E=(S_{2}-S_{1})\;T\;,$$
i.e., Landauer’s principle.

## 6. Irreversible Transport

Figure 7a represents a transport channel between two reservoirs. The upper reservoir has parameter values $T^{\prime }$, $\mu^{\prime }$, and $S^{\prime }$; the lower reservoir has parameter values $T^{\prime \prime }$, $\mu^{\prime \prime }$, and $S^{\prime \prime }$. We assume that the computer hardware is at a uniform temperature. Hence, $T^{\prime \prime }=T^{\prime }$ (say *T*). Furthermore, we assume that $S^{\prime \prime }=S^{\prime }$ (say *S*). This means, e.g., that noise does not cause random bit errors during the transport of the information. Thus, reservoirs only differ by $\mu^{\prime }$ and $\mu^{\prime \prime }$. See Figure 7b.

The particle drift *N* is caused by the difference of the two potentials $\mu^{\prime }$ and $\mu^{\prime \prime }$. The law governing the current is not necessarily linear. Hence, we have
$$N=\frac{1}{R}\;\left[\;f\left(\mu^{\prime }\right)-f\left(\mu^{\prime \prime }\right)\;\right]\;,$$
where f(μ) is an appropriate (monotonically increasing) function of μ and *R* (called resistance) is a constant depending primarily on the material properties and geometry of the particle channel. Many different functions *f* are applicable in different circumstances. For example, in classical c-MOS technology, an electron or hole diffusion process in silicon [14,15] can successfully be modeled by the function
$$f\left(\mu \right)=\exp\left(\frac{\mu }{kT}\right)\;,$$
such that
$$N=\frac{1}{R}\;\left[\;\exp\left(\frac{\mu^{\prime }}{kT}\right)-\exp\left(\frac{\mu^{\prime \prime }}{kT}\right)\;\right]\;.$$

Below, however, for sake of mathematical transparency, we will apply
f(μ)=μ,
such that we have a linear transport equation, i.e., Ohm’s law:$$N=\frac{1}{R}\;\left(\mu^{\prime }-\mu^{\prime \prime }\right)\;.$$

## 7. Endoreversible Engine

Figure 8 shows an endoreversible computer gate. It consists of
a core part with reversible gate andtwo transport channels: one for providing the input information and one for draining the output information.

The core is modeled according to Section 5; the two transport channels are modeled according to Section 6. In Figure 8a, the two outermost reservoirs (i.e., the input and output data registers) have fixed boundary conditions: $T_{1},\mu_{1},S_{1}$ and $T_{2},\mu_{2},S_{2}$, respectively. The inner parameters $T_{3}$, $\mu_{3}$, $S_{3}$, $T_{4}$, $\mu_{4}$, and $S_{4}$ are variables. In accordance with Section 6, we choose $T_{3}$ and $S_{3}$ equal to $T_{1}$ and $S_{1}$, respectively, as well as $T_{4}$ and $S_{4}$ equal to $T_{2}$ and $S_{2}$, respectively. Finally, we assume the whole engine is isothermal, such that $T_{2}=T_{1}$. This results in Figure 8b. We thus only hold back as variable parameters the chemical potentials $\mu_{3}$ and $\mu_{4}$.

According to Section 5, we have, for the core of the endoreversible engine:(5)$$E=\left(\mu_{3}-\mu_{4}\right)N+\left(S_{2}-S_{1}\right)T\;.$$

According to Section 6, the two transport laws are
(6)$$\begin{array}{ccc} N & = & \frac{1}{R_{1}}\;\left(\mu_{1}-\mu_{3}\right) \end{array}$$
(7)$$\begin{array}{ccc} N & = & \frac{1}{R_{2}}\;\left(\mu_{4}-\mu_{2}\right)\;. \end{array}$$

We remind that, in the present model, the intensive quantities *T*, $\mu_{1}$, $S_{1}$, $\mu_{2}$, and $S_{2}$ have given values, whereas the quantities $\mu_{3}$ and $\mu_{4}$ have variable values. We eliminate the two parameters $\mu_{3}$ and $\mu_{4}$ from the three Equations (Equation 5)–(Equation 7). We thus obtain
$$E\left(N\right)=\left(\mu -RN\right)\;N+T\left(S_{2}-S_{1}\right)\;,$$
where $\mu =\mu_{1}-\mu_{2}$ and $R=R_{1}+R_{2}$. The total energy dissipated in the endoreversible engine is
$$F=N\mu_{1}-E-N\mu_{2}\;.$$

We thus find
(8)$$F\left(N\right)=RN^{2}+T\left(S_{1}-S_{2}\right)\;.$$

The former term on the rhs of Equation (Equation 8) consists of the energy $R_{1}N^{2}$ dissipated in resistor $R_{1}$ and the energy $R_{2}N^{2}$ dissipated in resistor $R_{2}$; the latter term is the energy dissipated in the information loss in the core of the engine.

If N=0, then we dissipate a minimum of energy $F_{min}=F\left(0\right)=T\left(S_{1}-S_{2}\right)$. Unfortunately, N=0 corresponds with an engine computing infinitely slowly (just like a heat engine produces power with a maximum, i.e., Carnot, efficiency when driven infinitely slowly). For reasons of speed, a computer is usually operated in so-called short-circuit mode: $N=N_{sc}=\mu /R$. This operation corresponds with a short circuit between the two inner reservoirs of Figure 8: $\mu_{3}=\mu_{4}$. Under such condition, we have
(9)$$\begin{array}{ccc} F=F_{sc}=F\left(N_{sc}\right)=F\left(\frac{\mu }{R}\right) & = & \frac{1}{R}\;\mu^{2}+T\left(S_{1}-S_{2}\right)\\ & = & RN_{sc}^{2}+T\left(S_{1}-S_{2}\right)\\ & = & \mu N_{sc}+T\left(S_{1}-S_{2}\right)\;. \end{array}$$

For the sake of energy savings, we aim at low $F_{sc}$. As $N_{sc}$ is large for sake of speed, we thus need both small *R* and small μ.

## 8. Discussion

Figure 8b acts as our thermodynamical model of classical (i.e., non-quantum) computing. It is equally applicable to a single elementary logic gate as it is to a complete supercomputer. In the use of the model, we distinguish two cases: logically irreversible computation and logically reversible computation:

### 8.1. Conventional Computing

In conventional computers, information is erased during the computational process: $S_{1}-S_{2}>0$. The computation happens in one direction: from reservoir # 1 to reservoir # 2. In other words: N>0 and thus $\mu_{1}>\mu_{2}$.

During several decades, classical MOS technology succeeded in minimizing both parameters *R* and μ. Indeed, according to Moore’s law, we have known a continuous (exponential) shrinking of computer components. As a consequence, energy consumption per computational step diminished accordingly. In spite of this, the former term in (Equation 9) is still several orders of magnitude larger than the latter contribution. For a single computational step with e.g., a nand gate, in a 10 nanometer c-MOS technology, run with a power-supply voltage of 0.6 volt and operating at room temperature (i.e., at about 300 K), the dissipation in the resistor *R* is of the order of one attojoule (i.e., $10^{-18}$ J), whereas the dissipation $T(S_{1}-S_{2})$ is of order one zeptojoule (i.e., $10^{-21}$ J) only (corresponding with an entropy production of 3 zJ/K and 0.003 zJ/K, respectively). An attojoule may sound as an irrelevant miniscule amount of energy. However, because of present-day computer speeds (i.e., clock rates of about 3 GHz) and computation parallelism, it is responsible for the approximately 50 W dissipation in a single silicon chip (CPU or central processing unit) and hence for the megawatt power consumption in today’s data centres. Further improvement of technology could lead to less energy dissipation. However, breaking the Landauer barrier will be impossible without radically changing computation architecture, i.e., without switching from logically irreversible computing to reversible computing.

### 8.2. Reversible Computing

In reversible computers, no information is erased: $S_{1}-S_{2}=0$. The computation can happen in either direction: either from reservoir # 1 to reservoir # 2 or from reservoir # 2 to reservoir # 1. The former operation is activated by choosing $\mu_{1}>\mu_{2}$ (and hence N>0); the latter operation is activated by choosing $\mu_{2}>\mu_{1}$ (and hence N<0). For example, the circuits in Figure 1 indeed can be operated from top to bottom as well as from bottom to top, depending on the applied voltages.

Because the contribution $T(S_{1}-S_{2})$ to the dissipated energy *F* is absent from (Equation 8), we can, in the limit, make *F* as small as we like, by letting *N* go to zero, either from the positive side or from the negative side. This reminds us of the fundamental law: things only can happen without dissipation if they happen infinitely slowly.

## 9. Conclusions

Endoreversible schemes have proven to be very useful in many branches of science: both in thermodynamics and in many disciplines far beyond. In the present paper, we have applied it to informatics and computing. The simple model presented brings together Carnot’s law, Landauer’s principle, Ohm’s law, and even Moore’s law.

## Figures and Tables

**Figure 1 entropy-22-00660-f001:**
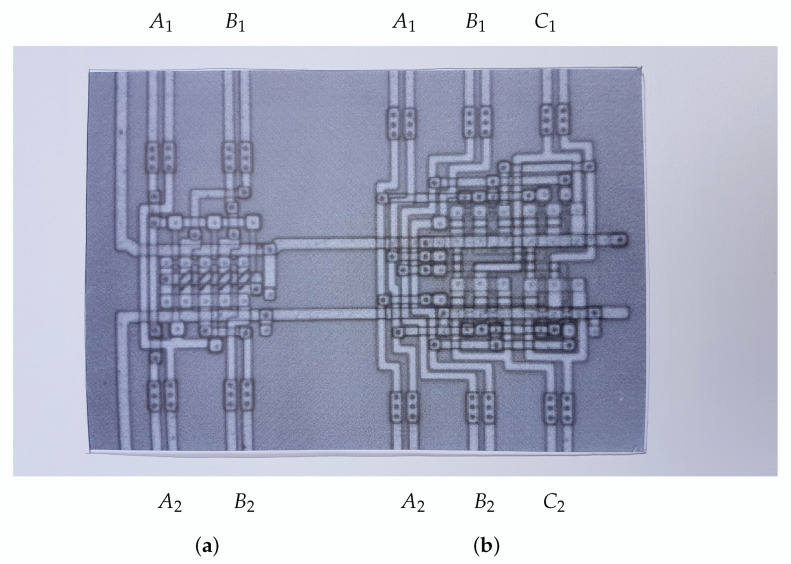
Silicon c-MOS inplementation of two reversible logic gates: (**a**) the controlled not gate and (**b**) the controlled controlled not gate.

**Figure 2 entropy-22-00660-f002:**
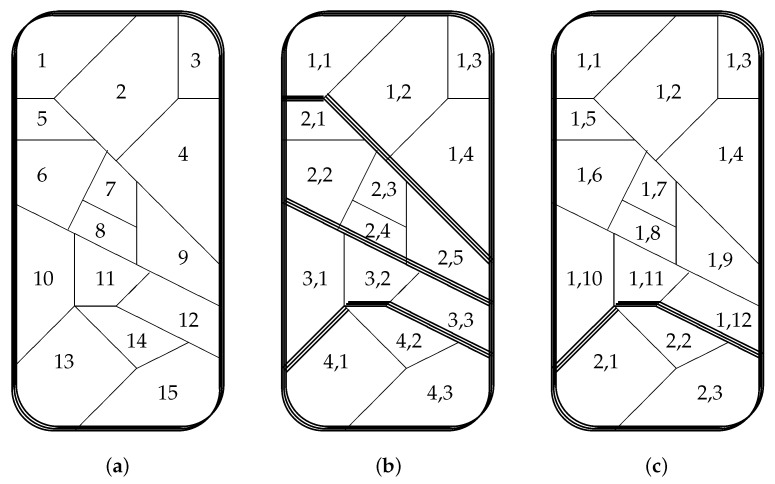
A same phase space divided into three different numbers *M* of macrocells: (**a**) M=1; (**b**) M=4; and (**c**) M=2.

**Figure 3 entropy-22-00660-f003:**
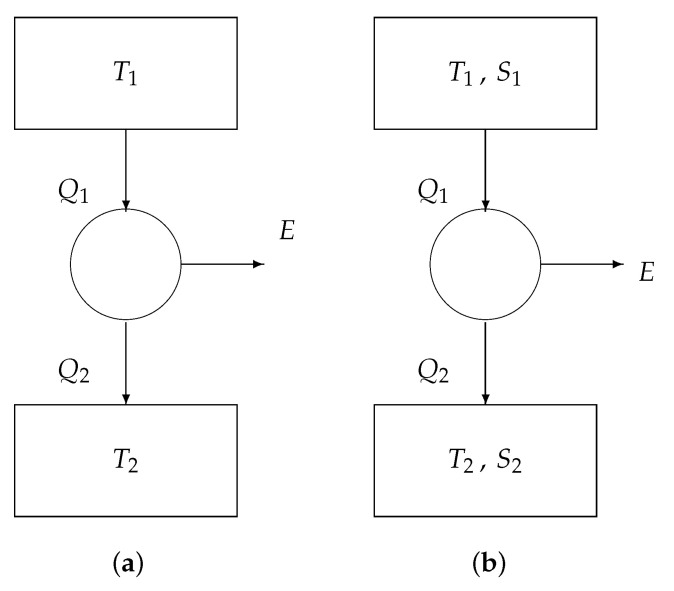
Core engines: (**a**) basic model and (**b**) extended model.

**Figure 4 entropy-22-00660-f004:**
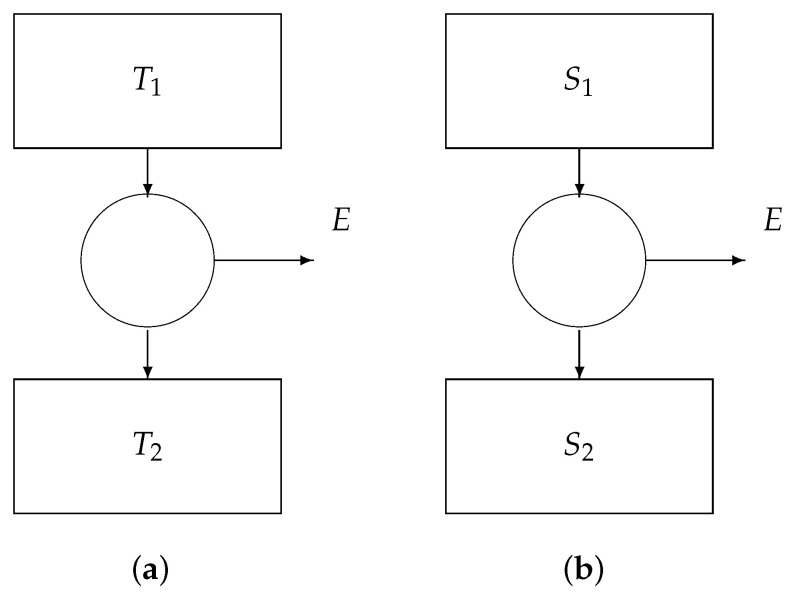
Core engines: (**a**) the Carnot engine and (**b**) the Landauer engine.

**Figure 5 entropy-22-00660-f005:**
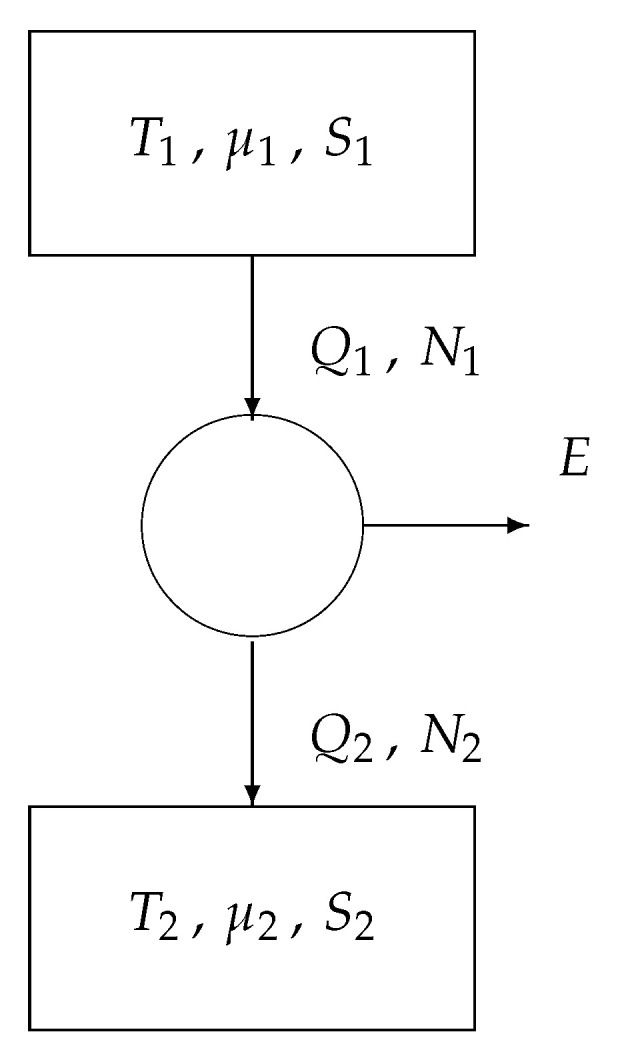
Core engine.

**Figure 6 entropy-22-00660-f006:**
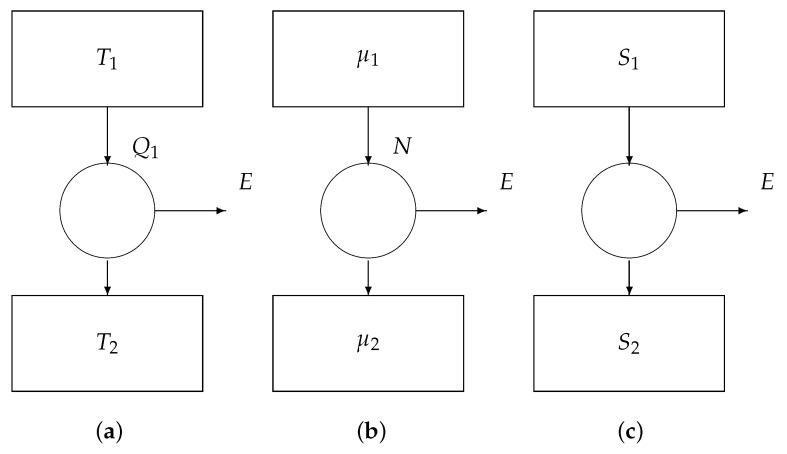
Core engines: (**a**) the Carnot engine, (**b**) the Gibbs engine, and (**c**) the Landauer engine.

**Figure 7 entropy-22-00660-f007:**
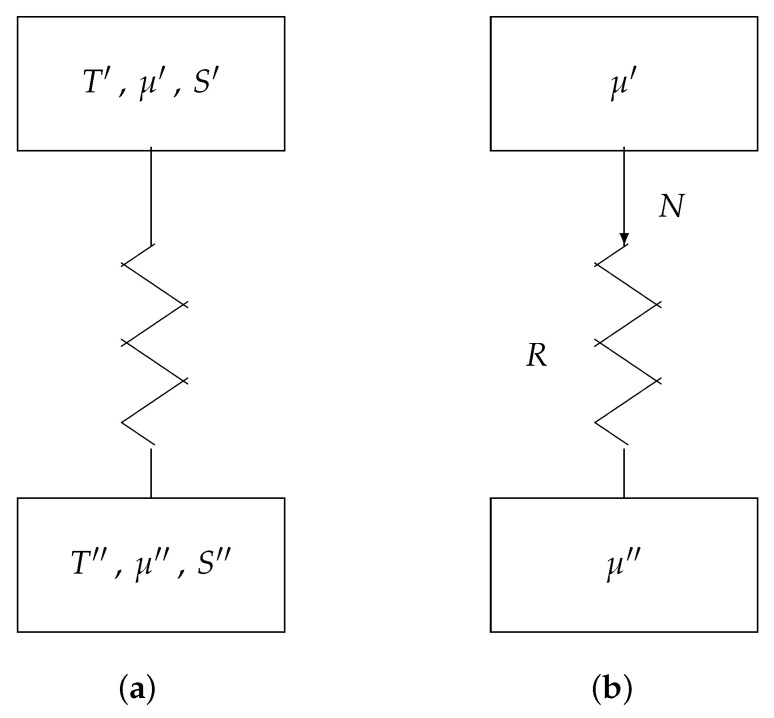
Irreversible transport: (**a**) general model and (**b**) simplified model.

**Figure 8 entropy-22-00660-f008:**
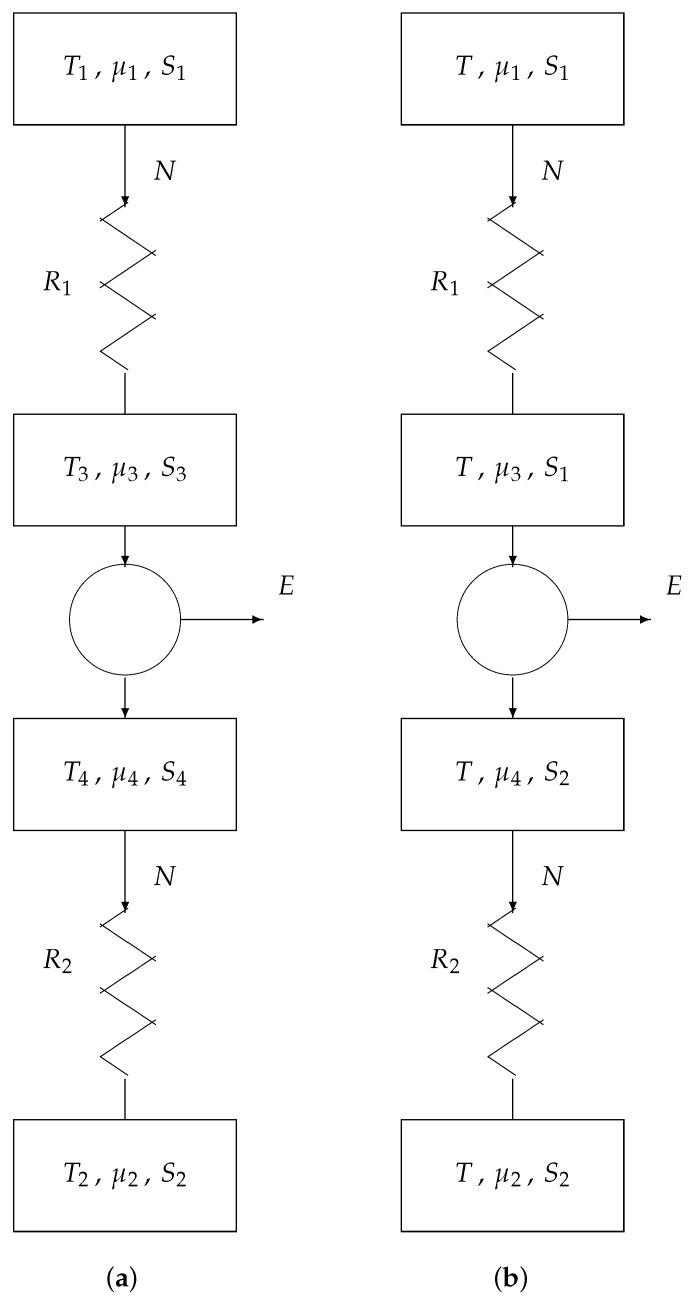
Endoreversible engine: (**a**) general model and (**b**) simplified model.

**Table 1 entropy-22-00660-t001:** Truth tables of two conventional logic gates: (**a**) the and gate and (**b**) the not gate.

(a)		
**A${}_{1}$**	**B${}_{1}$**	**A${}_{2}$**
0	0	0
0	1	0
1	0	0
1	1	1
(**b**)		
	**A${}_{1}$**	**A${}_{2}$**
	0	1
	1	0

**Table 2 entropy-22-00660-t002:** Truth tables of two reversible logic gates: (**a**) the controlled not gate and (**b**) the controlled controlled not gate.

(a)					
	**A${}_{1}$**	**B${}_{1}$**	**A${}_{2}$**	**B${}_{2}$**	
	0	0	0	0	
	0	1	0	1	
	1	0	1	1	
	1	1	1	0	
(**b**)					
**A${}_{1}$**	**B${}_{1}$**	**C${}_{1}$**	**A${}_{2}$**	**B${}_{2}$**	**C${}_{2}$**
0	0	0	0	0	0
0	0	1	0	0	1
0	1	0	0	1	0
0	1	1	0	1	1
1	0	0	1	0	0
1	0	1	1	0	1
1	1	0	1	1	1
1	1	1	1	1	0

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
