# Peer review of "Endoreversible Models for the Thermodynamics of Computing"

_entropy, 2020, doi:10.3390/e22060660_

Round 1

Reviewer 1 Report

The paper deals with a Finite Time Thermodynamic approach to the thermodynamics of computing. The paper addresses a standard model of computing where charged particles carry the information through the machine. (not a quantum computer). As in the endo reversible heat engine model temperature and chemical potential gaps are placed on the interface of the machine.to facilitate finite processing power. The study is a natural extension of the main ideas developed for heat engines. 

What is missing from this analysis is some numbers for example for C-MOS to quantify the amount of entropy production. 

Author Response

Manuscript entropy - 825788
Endoreversible models for the thermodynamics of computing

for the Entropy special issue on Finite-Time Thermodynamics

cover letter of revisions

(1) According to the wish of Reviewer # 1,
some numbers concerning the entropy production
in c-MOS are given in pages 16-17.

(2) According to the wish of Reviewer # 2,
a sentence after equation 3 (bottom of page 6)
has been expanded.

(3) The references have been reformatted
according to the MDPI guides.

I did not succeed in highlighting
these revisions in red color.
I apologize for the inconvenience.

Alexis De Vos
10 June 2020

Reviewer 2 Report

This is a very well written manuscript on the endoreversible modelling of information processing. It gives the reader a direct access into the thermodynamic and information-theoretic considerations for reversible and irreversible computing. The material is presented in a clear and structured fashion. I have only one remark for a potential improvement.

After eq. 3 one might expand the third sentence:

“We associate the microentropy with the heat Q, i.e. with that energy exchange, which would occur, if the microentropy enters or leaves the system at temperature T. According to . . . 

Author Response

(The authors gave the same response as above.)
